# Production, Mechanical Properties and Biomedical Characterization of ZrTi-Based Bulk Metallic Glasses in Comparison with 316L Stainless Steel and Ti6Al4V Alloy

**DOI:** 10.3390/ma15010252

**Published:** 2021-12-29

**Authors:** Mariusz Hasiak, Beata Sobieszczańska, Amadeusz Łaszcz, Michał Biały, Jacek Chęcmanowski, Tomasz Zatoński, Edyta Bożemska, Magdalena Wawrzyńska

**Affiliations:** 1Department of Mechanics, Materials and Biomedical Engineering, Wrocław University of Science and Technology, Smoluchowskiego 25, 50-370 Wroclaw, Poland; amadeusz.laszcz@pwr.edu.pl (A.Ł.); michal.bialy@pwr.edu.pl (M.B.); 2Department of Microbiology, Wroclaw Medical University, T. Chałubińskiego 4, 50-368 Wroclaw, Poland; beata.sobieszczanska@umw.edu.pl (B.S.); edyta.bozemska@umw.edu.pl (E.B.); 3Department of Advanced Materials Technologies, Wrocław University of Science and Technology, 25 Smoluchowskiego, 50-370 Wroclaw, Poland; jacek.checmanowski@pwr.edu.pl; 4Department and Clinic of Otolaryngology Head and Neck Surgery, Wroclaw Medical University, Borowska 213, 50-556 Wroclaw, Poland; tomasz.zatonski@umw.edu.pl; 5Center of Preclinical Studies, Wroclaw Medical University, Ludwika Pasteura 1, 50-367 Wroclaw, Poland; magdalena.wawrzynska@umw.edu.pl

**Keywords:** biocompatibility, bulk metallic glasses, Zr-based alloys, nanoindentation, fibroblasts, biocorrosion, cytotoxicity

## Abstract

Microstructure, mechanical properties, corrosion resistance, and biocompatibility were studied for rapidly cooled 3 mm rods of Zr_40_Ti_15_Cu_10_Ni_10_Be_25_, Zr_50_Ti_5_Cu_10_Ni_10_Be_25_, and Zr_40_Ti_15_Cu_10_Ni_5_Si_5_Be_25_ (at.%) alloys, as well as for the reference 316L stainless steel and Ti-based Ti6Al4V alloy. Microstructure investigations confirm that Zr-based bulk metallic samples exhibit a glassy structure with minor fractions of crystalline phases. The nanoindentation tests carried out for all investigated composite materials allowed us to determine the mechanical parameters of individual phases observed in the samples. The instrumental hardness and elastic to total deformation energy ratio for every single phase observed in the manufactured Zr-based materials are higher than for the reference materials (316L stainless steel and Ti6Al4V alloy). A scratch tester used to determine the wear behavior of manufactured samples and reference materials revealed the effect of microstructure on mechanical parameters such as residual depth, friction force, and coefficient of friction. Electrochemical investigations in simulated body fluid performed up to 120 h show better or comparable corrosion resistance of Zr-based bulk metallic glasses in comparison with 316L stainless steel and Ti6Al4V alloy. The fibroblasts viability studies confirm the good biocompatibility of the produced materials. All obtained results show that fabricated biocompatible Zr-based materials are promising candidates for biomedical implants that require enhanced mechanical properties.

## 1. Introduction

Metallic glasses belong to a relatively new group of advanced engineering materials, which exhibit enhanced mechanical properties [1,2,3,4,5,6,7,8,9,10]. The extraordinary mechanical parameters of these alloys are due to the amorphous internal structure, in contrast to classical metals and alloys with a crystalline structure. The manufacturing process of these materials requires a particular chemical composition [11,12,13,14,15] and high cooling rates [16,17,18], which can be obtained by a rapid solidification technique [18,19,20]. These production conditions provide sufficiently fast heat dissipation, restraining the possibility of reorganizing free atoms into the ordered crystal lattice. Today, due to the proper alteration of the chemical composition of modern bulk metallic glasses (BMGs) [21], various amorphous metallic alloys with dimensions of several centimeters can be produced [18,22,23].

The disordered internal structure of BMGs results in significant changes in physical properties, such as high hardness and strength [15,24,25,26,27,28,29,30], low elastic modulus [15,31,32], low macroscopic range of plastic deformations [33,34], and a high range of elastic deformations exceeding 2% [22,26,27], which subsequently leads to an increase in the elastic to total deformation energy ratio. From the point of view of potential application, the mechanical parameters of the metallic glasses mentioned above result in high resistance to scratching and abrasive wear [35,36,37,38,39].

One of the most distinctive features of BMGs that distinguishes them from traditional metallic alloys is the glass transition (*T_g_*), during which material in a supercooled liquid state transforms into a glassy structure upon cooling from high to low temperature. The glass transition is a reversible transformation because during heating, the glassy structure can again transform into the supercooled liquid state [30,40]. The temperature range between the glass transition temperature (*T_g_*) and the first crystallization temperature (*T_x_*) is called the supercooled liquid region (SLR) and is one of the main important parameters of BMGs [40,41,42]. The physicochemical properties of the materials change abruptly after entering the supercooled liquid region: viscosity drops [30], and large plastic deformations are allowed in the entire volume of the material [34,43]. This enables the thermoplastic processing of metallic glasses to form complex shapes using technologies similar to those used in the processing of thermoplastic polymers [18,41,44,45,46,47]. It is a very promising property that may extend the manufacturing possibilities of BMGs because nowadays, the requirement of high cooling rates restricts the forms and dimensions of currently fabricated metallic glasses. The ability to form metallic materials such as thermoplastic polymers offers attractive opportunities for biomedical applications, as the vast majority of BMGs are characterized by higher strength and elasticity than most currently available biometals.

Metallic amorphous alloys are also often characterized by excellent corrosion resistance. It is mainly related to the absence of grain boundaries, which are easy corrosion paths. Furthermore, the homogeneity of the chemical composition of BMGs (and often the content of many passivating elements) is also a significant advantage, which influences the corrosion resistance of the alloys in physiological fluids and various aggressive solutions [30,48,49,50]. The exceptional corrosion resistance of many glassy alloys may go hand in hand with the enhanced biocompatibility, associated with the ability to be inert in the organism’s environment and the lack of cytotoxicity of the released ions [51,52]. Alloys based on titanium and zirconium stand out in this field in particular [51,53,54,55,56,57,58,59,60,61].

The combination of favorable mechanical properties (i.e., high strength and low elastic modulus) and high wear resistance, as well as superior corrosion resistance in biological environments and enhanced biocompatibility, is very beneficial due to the use of metallic glasses for medical implants. The advantages of modern BMGs may overcome the limitations of currently used biomaterials. For example, one of the main disadvantages of popular 316L stainless steel is high elastic modulus, much higher than human bone. This discrepancy between the mechanical properties of bones and implants may lead to arthritis, osteoporosis, and other implant failures [62,63]. On the other hand, modern biocompatible Ti-based biometals are characterized by an elastic modulus close to the bone modulus. However, their biggest drawback is low hardness and, therefore, poor wear resistance [63,64,65]. 

The aim of this paper is the production of Zr-based bulk metallic glasses and their examinations of the microstructure to mechanical properties relationship, as well as biocorrosion and biocompatibility assessment. All investigations were performed in comparison to the reference 316L stainless steel and Ti-based Ti6Al4V alloy.

## 2. Materials and Methods

### 2.1. Production of Bulk Metallic Glasses

The Zr-based bulk metallic glasses (BMGs) with the nominal composition of Zr_40_Ti_15_Cu_10_Ni_10_Be_25_, Zr_50_Ti_5_Cu_10_Ni_10_Be_25_, and Zr_40_Ti_15_Cu_10_Ni_5_Si_5_Be_25_ (at.%) were prepared from an appropriate amount of high-purity elements under a protective Ar-atmosphere using an arc-melter with suction casting option [66,67,68]. Ingots were remelted and inverted five times to improve homogeneity. During the last melting process, the liquid alloys were sucked into the water-cooled copper mold, due to a negative pressure in the casting unit tanks. The high sucking force results in a quick movement of the molten alloy and causes rapid solidification of the material. Consequently, cylindrical rod samples with a diameter of 3 mm were manufactured. In order to compare the produced Zr-based BMGs with the commonly used biomedical metallic materials, the 316L surgical steel and Ti-based Ti6Al4V alloy were used in this work as reference materials [63,65].

### 2.2. Microstructure Investigations

Scanning electron microscope (SEM, FEI Quanta 250, FEI, Thermo Fisher Scientific, Waltham, MA, USA) working in backscattered electron mode (BSE) was used to observe the microstructures of the investigated materials. Additional energy-dispersive X-ray spectroscopy analysis (EDX) was carried out to confirm the chemical composition of the alloys. However, because of the EDX detection limits of light elements, at.% of Be atoms were locked to the constant level according to the expected chemical composition of Zr-based materials.

### 2.3. Investigations of Mechanical Properties

The mechanical properties of the prepared samples were studied using the Nanoindentation Tester (NHT2, CSM Instruments, Peseux, Switzerland) equipped with a pyramidal Berkovich tip. The load-controlled nanoindentation test was carried out with a loading rate of 400 mN/min to a maximum load of 200 mN, followed by a 10 s dwell at peak load and subsequent unloading with a rate of 400 mN/min. Analysis of the recorded load-displacement curves was performed according to the Oliver and Pharr protocol [69,70], allowing estimation of both plastic and elastic properties of the materials, including instrumented hardness (*H_IT_*) and elastic modulus (*E_IT_*), as well as plastic (*W_p_*) and elastic (*W_e_*) work during indentation. 

The tribological properties of the surface of the prepared Zr-based samples were investigated using a scratch tester (CSM Instruments, Micro-Scratch Tester, CSM Instruments, Peseux, Switzerland) with a standard Rockwell diamond tip (radius of 100 μm). All samples were ground with 400–2000 grit sandpaper and then polished with 6–0.25 μm diamond suspensions, followed by the final polishing step with 0.05 μm colloidal silica. Scratches of the length of 3 mm were recorded during a gradually increasing load from 0.5 to 10 N with a scratch speed of 1 mm/min. Pre- and post-scan measurements with the load of 0.03 N were also recorded to eliminate the potential surface slope and estimate the residual depth of the scratch after the test. The morphological character of the scratch was analyzed on a polarized optical microscope that was part of the scratch tester unit.

### 2.4. Corrosion Resistance in SBF

The DC electrochemical investigations were conducted in simulated body fluid (SBF) for 0.25, 24, 48, and 120 h of exposure time. Simulated body fluid was used as a corrosive environment, as it mimics the physiological environment comparable to that of human blood plasma. The SBF solution was prepared following the Kokubo methodology [71] (8.035 g NaCl, 0.355 g NaHCO_3_, 0.225 g KCl, 0.231 g K_2_HPO_4_ · 3H_2_O, 0.311 g MgCl_2_ · 6H_2_O, 39 mL 1.0 mol/L HCl, 0.292 g CaCl_2_, 0.072 g Na_2_SO_4_, 6.118 g (HOCH_2_)_3_CNH_2_ and 0–5 mL of 1.0 mol/L HCl). Electrochemical measurements were performed by recording polarization curves in a typical three-electrode configuration [72]. The measuring setup consisted of a measuring vessel and the Schlumberger SI 1286 potentiostat. The recordings started from the potential of –350 mV in the anode direction with a potential sweep rate of 1 mV/s. The potential was measured against a saturated calomel electrode (SCE). The corrosion current density (*i_corr_*) and polarization resistance (*R_p_*) were determined from the measured curves according to the Stern–Geary approach [73].

### 2.5. Biocompatibility Assay

#### 2.5.1. Cell Culture

The human fibroblast cell line IMR90 (ATCC CCL-186TM) was routinely cultured in Eagle’s Minimum Essential (EMEM) medium supplemented with 10% fetal bovine serum and 1% penicillin–streptomycin, and 1% nonessential amino acids (NEAA) solution. Cells were maintained in a humidified 5% CO_2_–95% air atmosphere at 37 °C. The culture medium was changed every second day.

#### 2.5.2. Cytotoxicity Assays

The direct cytotoxicity of bulk metallic alloys was evaluated according to the ISO 10993 standards [74]. In a direct cytotoxicity assay, IMR90 cells were seeded at a density of 1 × 105 cells/mL onto the polished and sterilized bulk metal alloy samples with a size of 2 mm × 2 mm × 1.5 mm in a 96-well plate. The 316L stainless steel and Ti6Al4V alloy samples were included for the comparative study. After 24 h, 48 h, and 72 h of incubation, the culture medium was discarded and replaced with the fresh medium containing 0.5 mg/mL MTT (3-(4,5-dimethylthiazol-2-yl)-2,5-diphenyltetrazolium bromide), which is metabolized by cellular mitochondrial dehydrogenases to purple-colored formazan crystals. The samples were incubated for 2 h at 37 °C, and then the formazan crystals were dissolved with DMSO. The absorbance of the formazan solution was detected spectrophotometrically at 570 nm. The results are reported as a percentage of cell viability compared to the negative control, that is, cells cultured without samples [75].

The indirect cytotoxicity was evaluated with extracts from the bulk metallic alloys in the cell culture medium, i.e., EMEM with all supplements. Samples of Zr-based alloys were incubated in a cell culture medium for 24 h, 72 h, and 120 h at 37 °C in a humidified 5% CO2–95% air atmosphere. The resulting extraction media were used to infect IMR90 cells cultured in a 96-well plate with 80% confluency. The negative control included untreated cells. Cell viability was evaluated with an MTT assay as described above [76].

#### 2.5.3. Microscopy

IMR90 cells that grow in the presence of bulk metallic alloys were inspected every day in the inverted and contrast-phase microscopes. Furthermore, to visualize cell morphology, after 24 h of incubation, cells grown on tested metallic alloys were fixed with 4% buffered formalin, washed three times with phosphate buffer, stained for 40 min with FITC-phalloidin (Sigma Aldrich, Merck Life Science, an affiliate of Merck KGaA, Darmstadt, Germany) to stain F-actin filaments, and with DAPI to stain cells nuclei.

#### 2.5.4. Statistical Analysis

Statistical analysis was performed on the mean of the data from three independent assays. The results are expressed as mean values (±SD). Analysis of variance was performed using one-way analysis of variance (ANOVA) with Tukey’s honest significance, test with *p* < 0.05 considered statistically significant.

## 3. Results and Discussion

### 3.1. Microstructure

It is well known that the enhanced physical properties of Zr-based bulk metallic glasses produced by the rapid cooling technique, compared to their classical counterparts, are due to their microstructure [26,27,77,78,79]. In Figure 1, the backscattered scanning electron microscopy (BSE-SEM) images with corresponding energy dispersive X-ray analysis (EDX) for all investigated bulk metallic glasses are presented. All images and chemical composition analyses were taken in the center of the sample. It is seen that the chemical compositions of the investigated alloys are in good agreement with the weighed amount of the individual components. The chemical composition of manufactured Zr-based alloys in SEM/EDX studies was determined for a fixed 25 at.% Be content of this alloying additive (Figure 1, right column). This assumption is caused by the fact that beryllium belongs to the group of light elements. The differences in chemical composition presented in Figure 1 are caused by the accuracy of the EDX method and the content of precipitates obtained in the production process selected for analysis (Figure 1, left column).

Microstructure images (Figure 1, left column) revealed the presence of small crystalline inserts, produced in the manufacturing process, embedded in an amorphous matrix. Three types of precipitates were observed in all studied alloys. The size, distribution, and number of crystallites depend on the chemical composition of the alloys. The highest number of inclusions was observed in the Si-containing sample, whereas the most uniform amorphous structure was found in the sample with the lowest Ti content.

### 3.2. Nanoindentation

It was shown in several research papers [15,24,25,26,27,28,29,30,79,80] that bulk metallic glasses exhibit excellent mechanical properties, such as high hardness and low Young’s modulus. In this paper, the mechanical properties were investigated on a microscale by applying nanoindentations tests. This approach allows one to determine the mechanical parameters for a single phase presented in the investigated samples.

In Figure 2a the instrumental hardness obtained for the produced Zr-based bulk metallic glasses, as well as for widely used 316L stainless steel and Ti6Al4V alloy, is presented. It is well seen that the instrumental hardness *H_IT_* for every single phase observed in the manufactured materials is higher than the hardness for the reference materials (316L stainless steel and Ti6Al4V alloy). Three phases in Zr-based alloys, denoted as phases A, B, and C, with increasing hardness were designated. The Zr_40_Ti_15_Cu_10_Ni_10_Be_25_ and Zr_50_Ti_5_Cu_10_Ni_10_Be_25_ samples, characterized by an almost completely amorphous structure (phase A), show hardness of 8.6 GPa and 8.0 GPa, respectively. The hardness of the precipitates (phase B and C) created in the production process ranges from 8.0 to 9.8 GPa. It is consistent with recent research reported in [81,82] about reduced atomic mobility, thus higher hardness, after the devitrification process. The Si-containing sample shows slightly higher values of hardness for each individual phase present in this alloy. It is connected with the diffusion process of silicon atoms from the amorphous matrix to the Si-rich phases. In this case, the hardness of the amorphous matrix (phase A) is about 9.1 GPa, whereas the hardness of crystallites marked as phases B and C is equal to 10.2 GPa and 10.8 GPa. For comparison, Figure 2a also shows the hardness for the 316L steel and Ti6Al4V alloy measured under the same conditions as for the Zr_40_Ti_15_Cu_10_Ni_10_Be_25_, Zr_50_Ti_5_Cu_10_Ni_10_Be_25_, and Zr_40_Ti_15_Cu_10_Ni_5_Si_5_Be_25_ alloys. It is seen that single-phase 316L stainless steel shows a hardness of about 3.7 GPa, which is more than two times less than that of the Zr-based alloys. Moreover, the two phases (phase α and phase β) of the Ti6Al4V alloy exhibit hardness of about 6 GPa for each phase present in the sample.

Figure 2b shows the plastic (*W_p_*), elastic (*W_e_*), and total (*W_t_*) energy dissipated during the indentation cycle for the Zr_40_Ti_15_Cu_10_Ni_10_Be_25_, Zr_50_Ti_5_Cu_10_Ni_10_Be_25_, Zr_40_Ti_15_Cu_10_Ni_5_Si_5_Be_25_, 316 stainless steel, and Ti6Al4V alloy. The depth–load course of penetration curves for Zr-based bulk metallic glasses investigated for a maximum load of 200 mN is comparable. Therefore, the values of hardness, as well as elastic, plastic, and total work, are similar. The elastic to total work ratio (*W_e_*/*W_t_*) for these materials is about 1/3, which is unusually high for typical metallic materials [29,83,84]. Quite a different behavior was observed for the 316L steel and Ti6Al4V alloy. The (*W_e_*/*W_t_*) values for the 316L steel and Ti6Al4V alloy are 1/10 and 1/4, respectively. For many biomedical applications, where a high *W_e_*/*W_t_* ratio is required, the Ti6Al4V alloy is more advantageous than 316L steel (Figure 2b) [63,85,86,87]. The results obtained for the manufactured Zr-based BMGs, summarized in Figure 2, show that they can be even more beneficial than both 316L steel and Ti6Al4V alloy and other typical crystalline biocompatible materials.

### 3.3. Scratch Test

The mechanical characterization of materials is one of the most significant studies in the context of the application of these materials in medicine. In addition to biocompatibility, these materials must satisfy several requirements, including wear resistance [87,88,89]. In Figure 3, the images of scratches and selected mechanical parameters for all the produced Zr-containing materials, as well as for the reference 316L steel and Ti6Al4V alloy recorded in scratch tests, are presented. It is well seen in Figure 3a that pictures of scratches for the 316L steel and Ti6Al4V alloy are distinctly different from the photos obtained for the Zr_40_Ti_15_Cu_10_Ni_10_Be_25_, Zr_50_Ti_5_Cu_10_Ni_10_Be_25,_ and Zr_40_Ti_15_Cu_10_Ni_5_Si_5_Be_25_ alloys. This difference is caused by the various hardness of the investigated materials and the different abilities for elastic deformation that result in subsequent elastic recovery. The reference materials exhibit noticeably deeper scratches with specific jagged edges. It is especially evident for the 316L steel. The characteristic of penetration depth (*P_d_* versus *L*) described by the slope angle of this curve (Figure 3b) is higher and noticeably different for the 316L steel than for the other samples. More significant changes for the reference materials compared to fabricated BMGs samples can be seen in the plot showing the residual depth versus scratch length (*R_d_* versus *L*) (Figure 3c). The residual depth (*R_d_*) for the 316L steel is approximately two times higher than for the Ti6Al4V alloy. Moreover, the Zr-containing materials have relatively low values of *R_d_* compared to those of the reference materials. The sudden decline in both penetration and residual depth plots is related to the microcracking that occurs in metallic glasses samples [90].

The changes in friction force (*F_t_*) and coefficient of friction (*μ*) (Figure 3c,d, respectively) recorded for all investigated materials are well correlated with each other, and they are reflected in the scratches images shown in Figure 3a. The rapid increase in *F_t_* and *μ* observed in these curves is related to material buildup during scratching. The peaks on the discussed plots for BMGs are connected with the microcracking of the material mentioned above. Visible change in friction force and coefficient of friction after about 0.6 mm for the Ti6Al4V alloy can be coupled with a change from sliding movement with mainly adhesion part of friction to ploughing friction [91]. It is also visible as a change in the nature of the scratch in Figure 3a.

### 3.4. Corrosion

The electrochemical investigations were carried out the manufactured Zr_40_Ti_15_Cu_10_Ni_10_Be_25_, Zr_50_Ti_5_Cu_10_Ni_10_Be_25_, and Zr_40_Ti_15_Cu_10_Ni_5_Si_5_Be_25_ materials, as well as reference 316L surgical steel and Ti6Al4V alloy. Figure 4 shows the potentiodynamic polarization curves recorded for the investigated and reference samples after different times of exposure to SBF.

After a short 15 min exposition in SBF (the so-called “initial state”), the corrosion current density is lower for the Zr-based bulk metallic glasses than for the reference 316L steel and Ti6Al4V alloy (Figure 4a, Table 1). After this short time of exposure, simulating the first contact of the material with the physiological environment of the human body, the lowest corrosion rates of approximately 8–9 × 10^−8^ A/cm^2^ were observed for the Zr-based alloys without the addition of Si. The corrosion rates of these alloys were 7.5 and 3 times lower than those for the 316L steel and Ti6Al4V alloy, respectively (Table 1). For the Si-containing sample, the corrosion rate was noticeably higher than that for the other Zr-based materials. However, the *i_corr_* was still 1.5 times lower for the Zr_40_Ti_15_Cu_10_Ni_5_Si_5_Be_25_ alloy in comparison to the reference 316L surgical steel, but unfortunately also 1.5 times higher than for the Ti6Al4V alloy. 

After 24 h of exposure to SBF (Figure 4b), biocompatibility (measured as high values of *R_p_*) [92] and the corrosion resistance (low *i_corr_*) of the Zr-based alloys increased compared to the reference 316L steel and the Ti6Al4V alloy. Significant changes in *R_p_* and *i_corr_* are strongly connected with the chemical composition of the alloys. The highest increase in *R_p_* with a simultaneous improvement in corrosion resistance was observed for the sample with the highest Zr content and the lowest Ti content (Zr_50_Ti_5_Cu_10_Ni_10_Be_25_ alloy). The corrosion resistance of this material is more than 39 times higher in comparison to the 316L steel, and almost 12 times higher in comparison to the Ti6Al4V alloy. The decrease in Zr content (to 40 at.%) and the increase in Ti content (to 15 at.%) in the Zr_40_Ti_15_Cu_10_Ni_10_Be_25_ alloy led to a threefold decrease in corrosion resistance compared to the Zr_50_Ti_5_Cu_10_Ni_10_Be_25_ alloy. However, the Zr_40_Ti_15_Cu_10_Ni_10_Be_25_ sample still exhibits 11 and 3.5 times higher *i_corr_* than the 316L steel and Ti6Al4V alloy, respectively (Table 1). The introduction of 5 at.% of Si at the expense of Ni to the ZrTiCuNiBe-based composition also changes the corrosion behavior of the material. For the Zr_40_Ti_15_Cu_10_Ni_5_Si_5_Be_25_ alloy, the corrosion resistance is located between that of Zr_50_Ti_5_Cu_10_Ni_10_Be_25_ and Zr_40_Ti_15_Cu_10_Ni_10_Be_25_ alloys. Very similar behavior of *R_p_* and *i_corr_* for the produced Zr-based materials and reference samples are observed after 48 h of exposition in the physiological environment (Figure 4c, Table 1). 

The extension of the exposure duration in SBF to 120 h for the investigated samples results in stabilizing the parameters of polarization resistance and corrosion current density parameters (Figure 4d, Table 1). Variations in these parameters decrease significantly after five days of exposure to the physiological environment. During the 120 h exposition, Zr-based bulk metallic glasses showed higher polarization resistance (*R_p_*) and lower corrosion rate (*i_corr_*) than the reference samples. A slight increase in *R_p_*, followed by a decrease in *i_corr_*, is observed in the produced materials (Table 1). The best corrosion resistance (*i_corr_*) and the highest possible biocompatibility (highest *R_p_*) after 5 days of exposure to SBF were again recorded for the Zr_50_Ti_5_Cu_10_Ni_10_Be_25_ and Zr_40_Ti_15_Cu_10_Ni_10_Be_25_ alloys. It is essential that all produced Zr-based bulk metallic glasses are characterized by better corrosion resistance than the reference samples. The exact values of the electrochemical corrosion parameters are summarized in Table 1. In general, it can be pointed out that Zr-based metallic glasses have excellent corrosion resistance and perform better than the 316L steel and Ti6Al4V alloy, which is also reported in other papers for various corrosion environments [60,61,93,94,95,96].

Another essential aspect of corrosion problems is the repassivation process of the outer surface of metallic materials. Potentiodynamic polarization curves recorded for the produced Zr-based materials after different exposure times in SBF indicate the noticeable changes in the rate of electrode processes taking place on the surface of the investigated samples (Figure 4). Both Zr and Ti, present in the chemical composition of the produced alloys, are easily passivated metals, resulting in the formation of a thin layer of ZrO_2_ or TiO_2_ (usually 10–20 nm thick) [49,60,94,97]. In a water corrosion environment containing chlorides, such as the SBF used in the presented study, the passive layers are subjected to the aggressive action of chloride ions, which may lead to a local breakdown of the passive film and possible subsequent repassivation. These electrochemical processes are observable as sudden changes in current density in both cathodic and anodic areas, which is visible in Figure 4. For the produced Zr-based materials, the repassivation processes are present during the entire duration of exposure to SBF, starting from the initial state (Figure 4a) and ending after 120 h (Figure 4d). From the very beginning of exposure to SBF (Figure 4a), Zr-based BMGs are characterized by a lower current density in the cathodic and anodic areas than the reference 316L surgical steel and Ti6Al4V alloy. These changes in electrochemical parameters show that the rate of electrode processes is slower on the surface of the Zr-based metallic glasses than on the reference samples. In addition, the differences in the values of the cathodic to anodic transition potential (*E_C-A_*) and open-circuit potential (*E_OC_*) of the produced materials show that during exposition to the physiological environment, the passive layer is prone to repassivation, which is the desired sign of a “self-healing” tendency. Significantly, similar electrochemical processes are also recorded for the 316L surgical steel and Ti6Al4V alloy, which is reflected in the values of *E_C-A_* and *E_OC_* parameters (Table 1).

### 3.5. Direct Cytotoxicity to Fibroblasts

Figure 5 shows the relative viability and proliferation of fibroblasts (IMR90 cell line) in direct contact with the investigated Zr-based BMGs and reference samples. During 24 h of incubation, the viability of fibroblasts cultured in the presence of Zr_40_Ti_15_Cu_10_Ni_10_Be_25_, Zr_50_Ti_5_Cu_10_Ni_10_Be_25_, and Zr_40_Ti_15_Cu_10_Ni_5_Si_5_Be_25_ samples decreased by 10.5% to 12.1% compared to the negative control group (NC). However, the cytotoxicity of the 316L stainless steel to fibroblasts was higher than for all Zr-based alloys. After incubation for 48 h and 72 h, the Zr_40_Ti_15_Cu_10_Ni_10_Be_25_, Zr_50_Ti_5_Cu_10_Ni_10_Be_25_, and Zr_40_Ti_15_Cu_10_Ni_5_Si_5_Be_25_ bulk metallic glasses showed the cytotoxicity to fibroblasts comparable to that determined for the 316L stainless steel and Ti6Al4V alloys.

Every day, the cell morphology was assessed by phase contrast microscopy. Figure 6 shows example images of the morphology of fibroblasts growing in the presence of the produced Zr-based BMGs compared to the negative control group, i.e., cells growing on glass slides. The cells growing near the alloy coupons were well anchored and presented a standard spindle morphology typical of unstimulated cells. The proper cell morphology indicates no or negligible cytotoxicity of the tested materials (Figure 6). However, fluorescence staining of cells’ actin cytoskeleton and nuclei revealed that cells growing on coupons with Si-containing material (Zr_40_Ti_15_Cu_10_Ni_5_Si_5_Be_25_ alloy) had more rounded morphology compared to the 316L stainless steel and Ti6Al4V alloy (Figure 7). These cells also demonstrated accumulation of actin at the edges of the cells. Similar fibroblasts morphology was also observed in all samples of the Ti6Al4V alloy. In general, fibroblasts cultured on all samples investigated in this paper demonstrated a less-stretched morphology than the control on the glass slide. Apparently, the hardness of the glass could influence the morphology of the fibroblasts (Figure 7).

### 3.6. Indirect Cytotoxicity to Fibroblasts

The relative viability of fibroblasts (IMR90 cell line) in extraction media for the investigated Zr-based BMGs and the reference 316L stainless steel and Ti6Al4V alloy is shown in Figure 8. A 24 h exposure of fibroblasts to the extraction media revealed no toxicity to fibroblasts. The produced Zr_40_Ti_15_Cu_10_Ni_10_Be_25_, Zr_50_Ti_5_Cu_10_Ni_10_Be_25_, and Zr_40_Ti_15_Cu_10_Ni_5_Si_5_Be_25_ metallic glasses and reference samples seemed to be inert in their bulk form to fibroblasts.

In conclusion, all manufactured Zr-based metallic glasses showed cytotoxicity to fibroblasts lower than 13% on the first day of incubation. However, the toxicity decreased to less than 7% on the third day of incubation, indicating the excellent compatibility of the tested alloys. Generally, the toxicity to fibroblasts for the all studied samples was similar to that of the 316L stainless steel and Ti6Al4V alloy. There were statistically insignificant differences (*p* = 0.999) between the fabricated Zr-based samples, 316L stainless steel, and Ti6Al4V alloy. 

Both studies indicate that all tested BMGs present excellent biocompatibility at a similar or even better level than the 316L steel and Ti6Al4V alloy. Biocompatibility is high despite the presence of potentially toxic nickel and beryllium. Such results were also confirmed by previous research [54] on the same group of materials. Other investigations on Zr-based BMGs with Ni content also showed their low cytotoxicity in normal conditions, due to their excellent passivation ability [98,99]. Moreover, it was also reported that, in terms of cytotoxicity, the upper safe level of nickel can reach even about 50% [100] in crystalline alloys, which is much more than in tested BMG samples. 

## 4. Conclusions

In this paper, the biocompatibility of the produced Zr_40_Ti_15_Cu_10_Ni_10_Be_25_, Zr_50_Ti_5_Cu_10_Ni_10_Be_25_, and Zr_40_Ti_15_Cu_10_Ni_5_Si_5_Be_25_ bulk metallic glasses were investigated in comparison to the commonly used 316L stainless steel and Ti6Al4V alloy. Biocompatibility studies were focused on the relationship between microstructure, mechanical parameters, and scratch resistance, as well as corrosion resistance in simulated body fluid and cytotoxicity evaluation followed by cell morphology description. The presented results show that materials produced by the rapid quenching technique exhibit a composite glassy nature with excellent mechanical properties. It is worth emphasizing that manufactured Zr-based materials exhibit hardness exceeding 8 GPa and exceptional scratch resistance (defined as low residual depth and high elastic to total energy ratio). The electrochemical investigations carried out in SBF up to 120 h show high biocorrosion resistance of fabricated BMGs, several times better than the reference 316L stainless steel and Ti6Al4V alloy. Furthermore, the investigations of the fibroblasts viability for the Zr_40_Ti_15_Cu_10_Ni_10_Be_25_, Zr_50_Ti_5_Cu_10_Ni_10_Be_25_, and Zr_40_Ti_15_Cu_10_Ni_5_Si_5_Be_25_ alloys present excellent biocompatibility and have great potential in medical applications, e.g., stents, artificial joints, and prostheses production, similar to the widely used 316L stainless steel and Ti6Al4V alloy.

## Figures and Tables

**Figure 1 materials-15-00252-f001:**
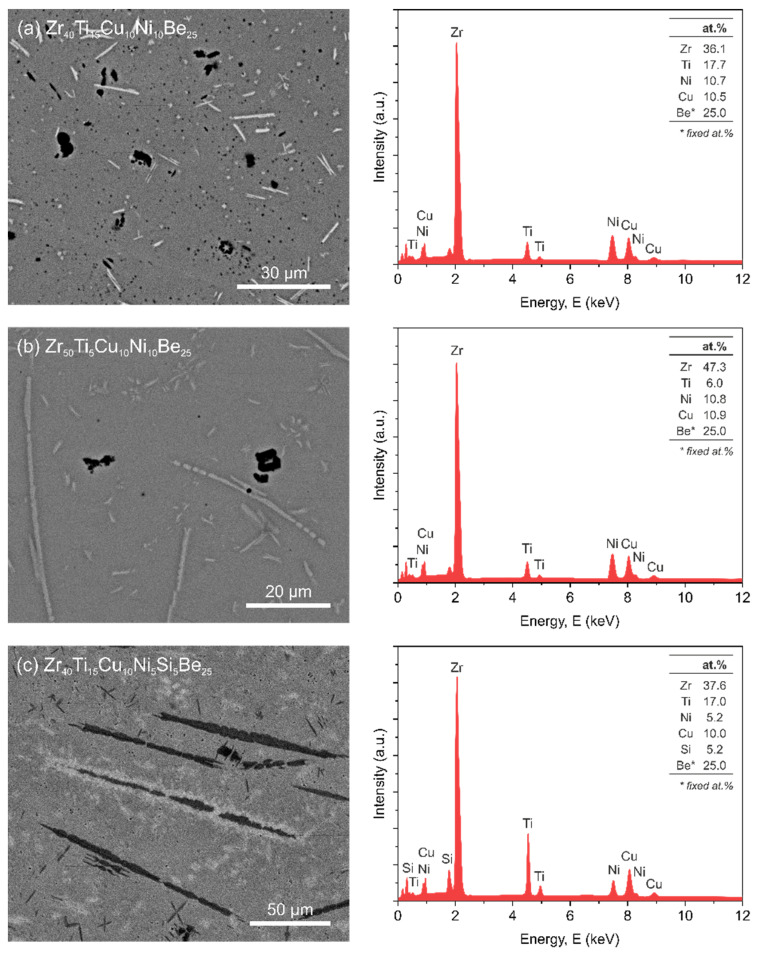
BSE SEM images (**left column**) and corresponding EDX analyses (**right column**) for the manufactured Zr_40_Ti_15_Cu_10_Ni_10_Be_25_, Zr_50_Ti_5_Cu_10_Ni_10_Be_25_, and Zr_40_Ti_15_Cu_10_Ni_5_Si_5_Be_25_ alloys.

**Figure 2 materials-15-00252-f002:**
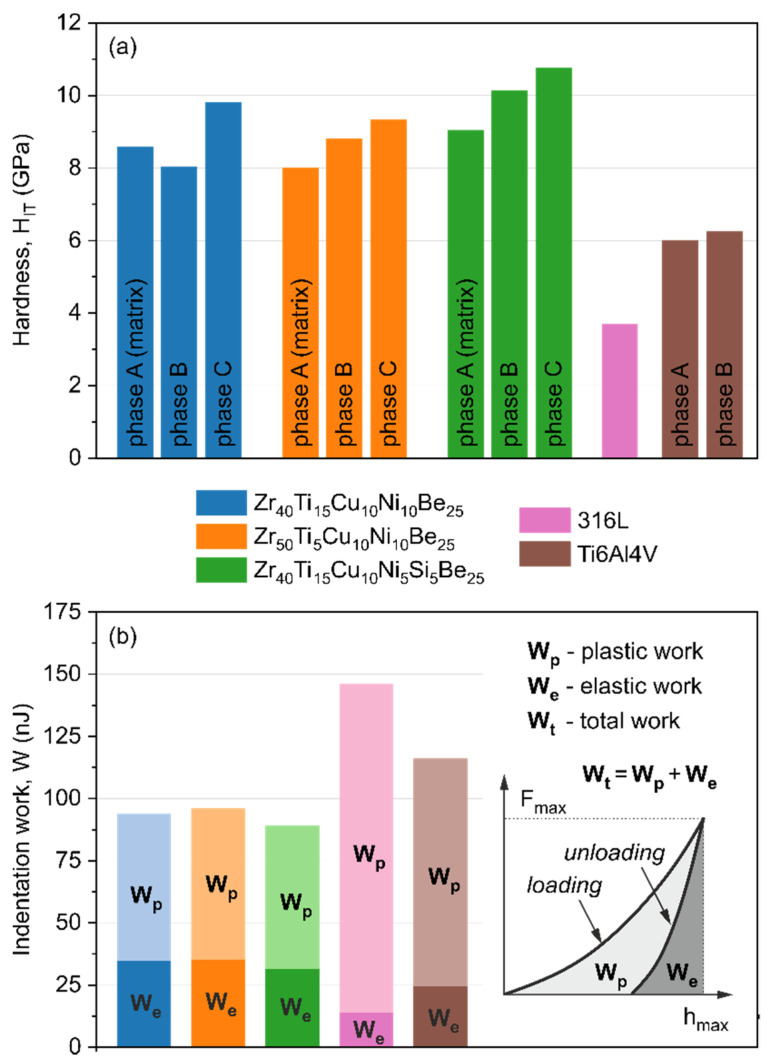
Bar plot representing the hardness of different phases observed in the materials (**a**) and the deformation energies (**b**) obtained in nanoindentation tests for the Zr_40_Ti_15_Cu_10_Ni_10_Be_25_, Zr_50_Ti_5_Cu_10_Ni_10_Be_25_, and Zr_40_Ti_15_Cu_10_Ni_5_Si_5_Be_25_ alloys, as well as reference 316L stainless steel and Ti6Al4V alloy.

**Figure 3 materials-15-00252-f003:**
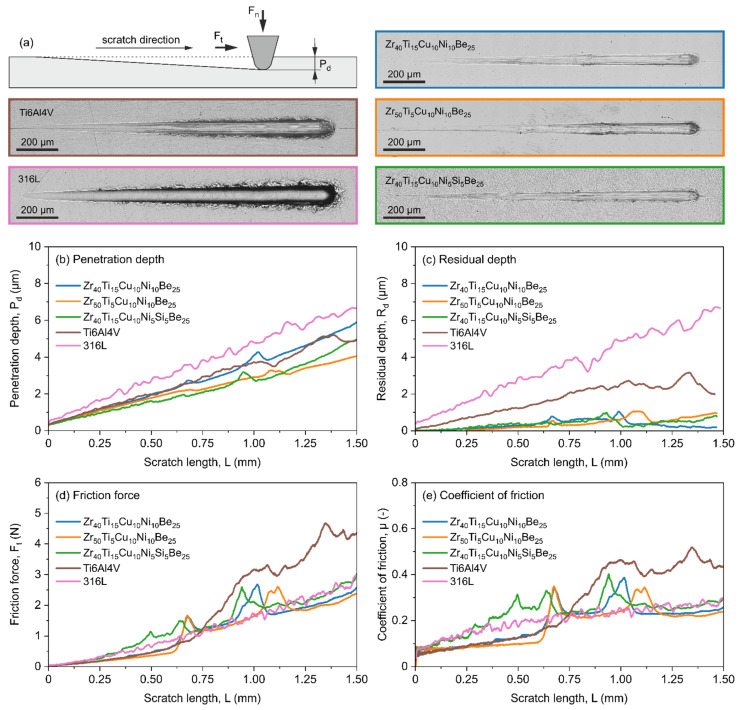
Photos of scratches (**a**) and mechanical parameters such as penetration depth (**b**), residual depth (**c**), friction force (**d**), and coefficient of friction (**e**) for the Zr_40_Ti_15_Cu_10_Ni_10_Be_25_, Zr_50_Ti_5_Cu_10_Ni_10_Be_25_, Zr_40_Ti_15_Cu_10_Ni_5_Si_5_Be_25_, and reference 316L surgical steel and Ti6Al4V alloy obtained in scratch tests.

**Figure 4 materials-15-00252-f004:**
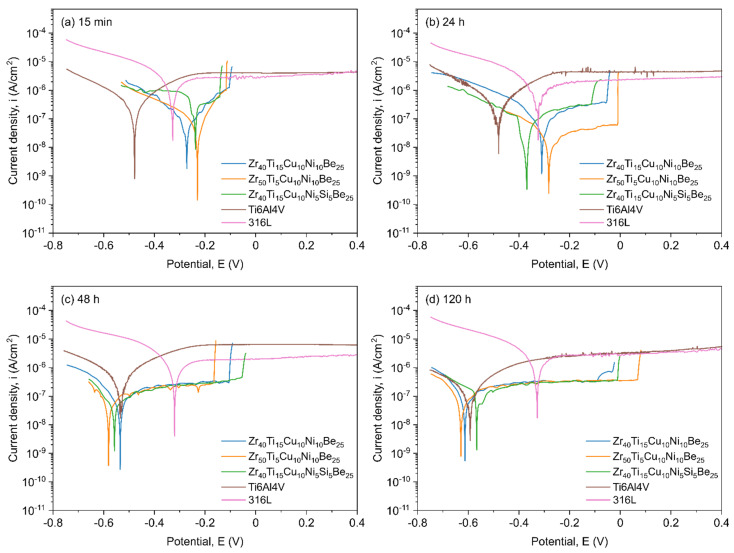
Potentiodynamic polarization curves for the produced Zr_40_Ti_15_Cu_10_Ni_10_Be_25_, Zr_50_Ti_5_Cu_10_Ni_10_Be_25_, and Zr_40_Ti_15_Cu_10_Ni_5_Si_5_Be_25_ alloys, reference 316L steel, and Ti6Al4V alloy recorded after different times of exposition in SBF: (**a**) 15 min, (**b**) 24 h, (**c**) 48 h, and (**d**) 120 h.

**Figure 5 materials-15-00252-f005:**
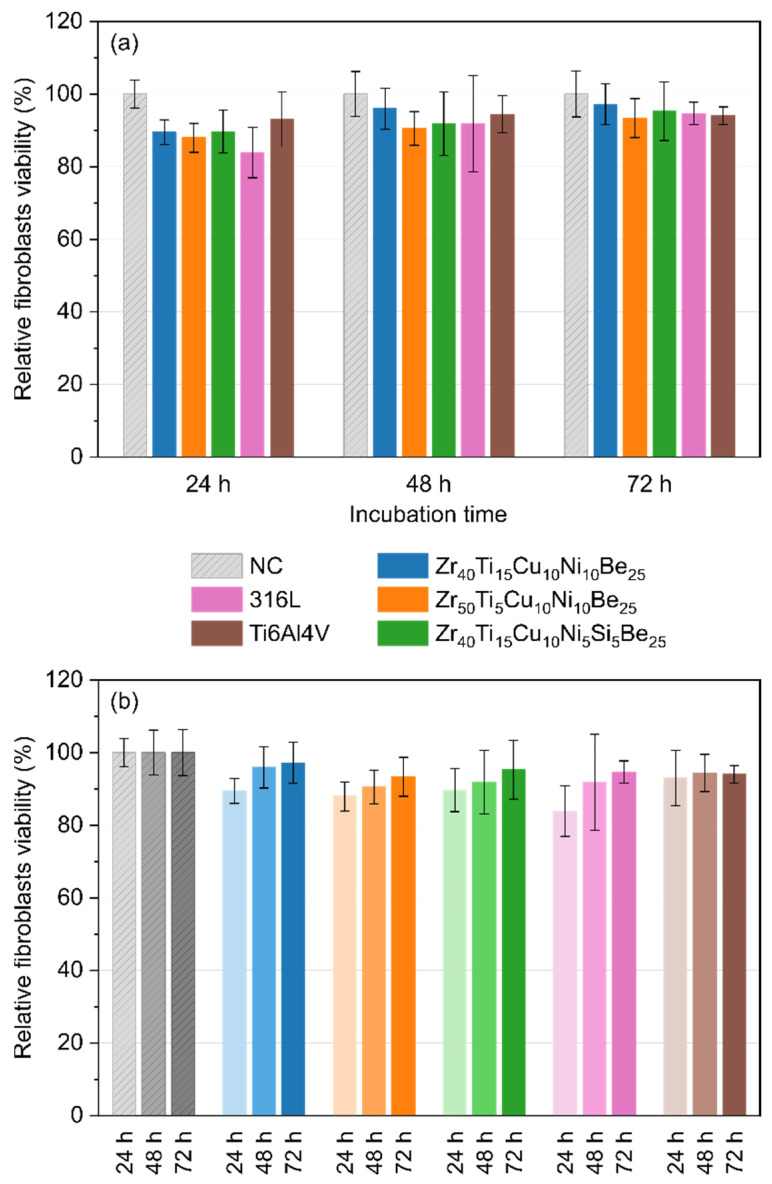
Relative viability and proliferation of fibroblasts (IMR90 cell line) in direct contact with Zr_40_Ti_15_Cu_10_Ni_10_Be_25_, Zr_50_Ti_5_Cu_10_Ni_10_Be_25_, Zr_40_Ti_15_Cu_10_Ni_5_Si_5_Be_25_ alloys, 316L stainless steel, and Ti6Al4V alloy assessed by MTT assay carried out at 24 h, 48 h, and 72 h time intervals—compatison between samples at the same incubation time (**a**), and influence of incubation time for each material (**b**). Cell viability of untreated cells (NC) was taken as 100%.

**Figure 6 materials-15-00252-f006:**
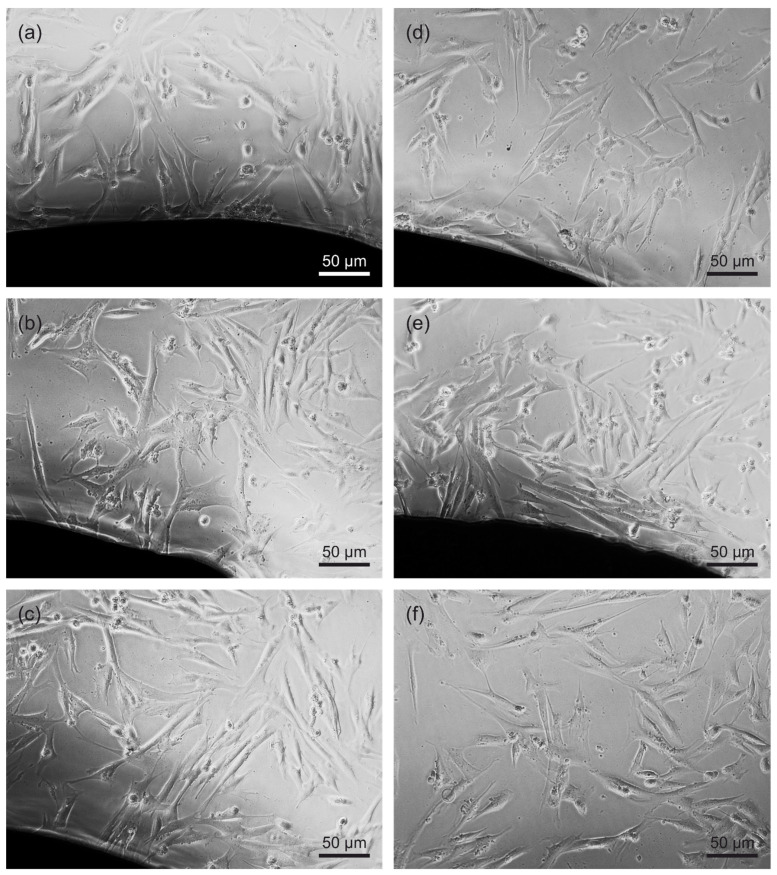
Phase contrast micrographs of fibroblasts (IMR90 cell line) cultured for 24 h on (**a**) Zr_40_Ti_15_Cu_10_Ni_10_Be_25_, (**b**) Zr_50_Ti_5_Cu_10_Ni_10_Be_25_, (**c**) Zr_40_Ti_15_Cu_10_Ni_5_Si_5_Be_25_, (**d**) 316L steel, and (**e**) Ti6Al4V alloy. (**f**) Fibroblasts cultured on glass slides served as a negative control (NC).

**Figure 7 materials-15-00252-f007:**
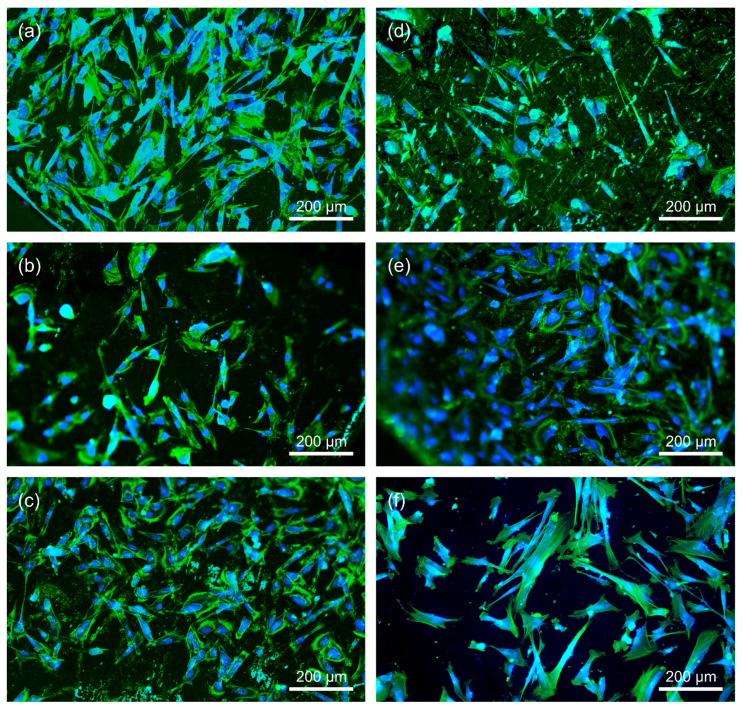
Fluorescence staining of the actin cytoskeleton of fibroblasts (IMR90 cell line) cultured for 24 h on (**a**) Zr_40_Ti_15_Cu_10_Ni_10_Be_25_, (**b**) Zr_50_Ti_5_Cu_10_Ni_10_Be_25_, (**c**) Zr_40_Ti_15_Cu_10_Ni_5_Si_5_Be_25_, (**d**) 316L steel, and (**e**) Ti6Al4V alloy. (**f**) Fibroblasts cultured on glass slides served as a negative control (NC).

**Figure 8 materials-15-00252-f008:**
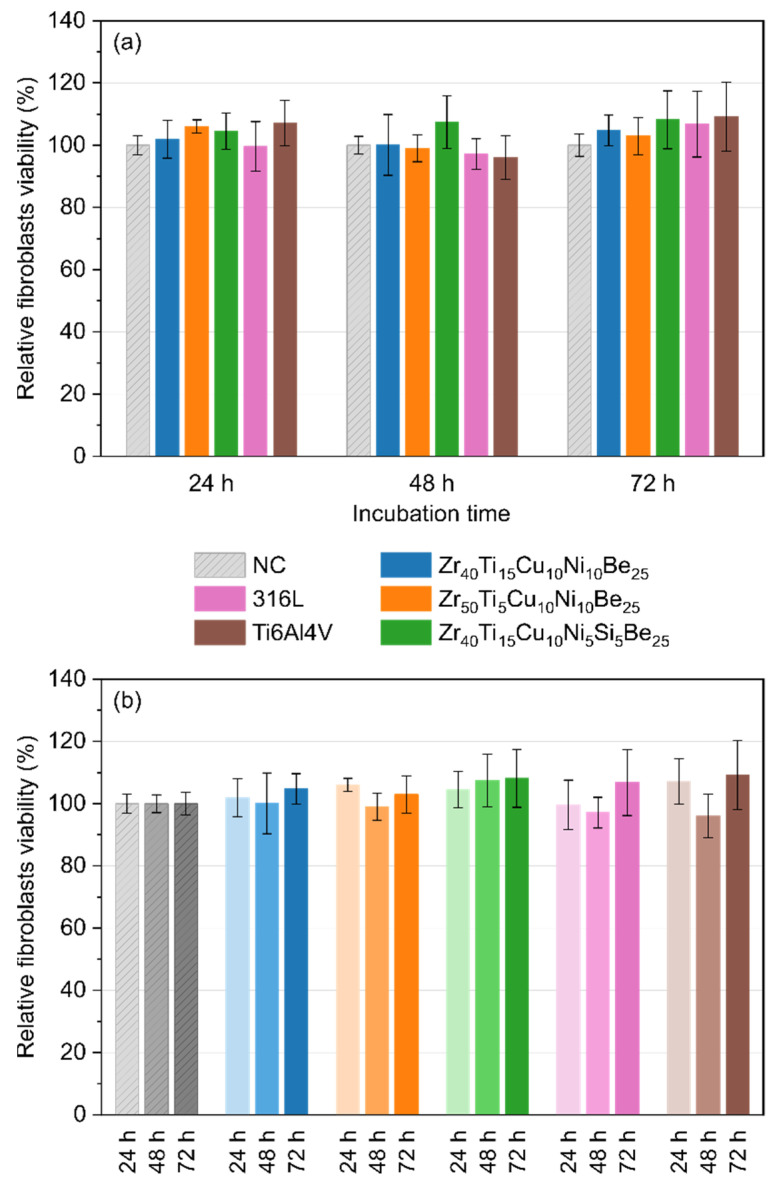
Relative viability of fibroblasts (IMR90 cell line) in extraction media from Zr_40_Ti_15_Cu_10_Ni_10_Be_25_, Zr_50_Ti_5_Cu_10_Ni_10_Be_25_, Zr_40_Ti_15_Cu_10_Ni_5_Si_5_Be_25_ alloys, 316L stainless steel, and Ti6Al4V alloy evaluated by MTT assay carried out at 24 h, 48 h, and 72 h time intervals—compatison between samples at the same incubation time (**a**), and influence of incubation time for each material (**b**). Cell viability of untreated cells (NC) was considered to be 100%.

**Table 1 materials-15-00252-t001:** Polarization resistance (*R_p_*), corrosion current density (*i_corr_*), cathodic to anodic transition potential (*E_C-A_*), and open circuit potential (*E_OC_*) for the produced Zr_40_Ti_15_Cu_10_Ni_10_Be_25_, Zr_50_Ti_5_Cu_10_Ni_10_Be_25_, Zr_40_Ti_15_Cu_10_Ni_5_Si_5_Be_25_ alloys, reference 316L steel, and Ti6Al4V alloy recorded after different times (0.25, 24, 48, and 120 h) of exposition in SBF.

Alloy Composition	Time (h)	*R_p_* (Ω·cm^2^)	*i_corr_* (A/cm^2^)	*E_C-A_* (mV)	*E_OC_* (mV)
Zr_40_Ti_15_Cu_10_Ni_10_Be_25_	0.25	2.77 × 10^5^	9.42 × 10^−8^	−274	−167
	24	3.42 × 10^5^	7.62 × 10^−8^	−308	−318
	48	3.09 × 105	8.44 × 10^−8^	−535	−348
	120	2.27 × 10^5^	1.15 × 10^−8^	−612	−123
Zr_50_Ti_5_Cu_10_Ni_10_Be_25_	0.25	3.24 × 10^5^	8.06 × 10^−8^	−229	−185
	24	1.18 × 10^5^	2.22 × 10^−8^	−280	−352
	48	6.99 × 10^4^	3.73 × 10^−7^	−633	−154
	120	2.72 × 10^5^	9.81 × 10^−8^	−627	−91
Zr_40_Ti_15_Cu_10_Ni_5_Si_5_Be_25_	0.25	6.32 × 10^4^	4.12 × 10^−7^	−240	−187
	24	6.27 × 10^5^	4.16 × 10^−8^	−373	−267
	48	3.21 × 10^5^	8.13 × 10^−8^	−558	−299
	120	1.49 × 10^5^	1.70 × 10^−7^	−562	−125
Ti6Al4V	0.25	9.91 × 10^4^	2.63 × 10^−7^	−478	−472
	24	9.79 × 10^4^	2.66 × 10^−7^	−483	−487
	48	8.05 × 10^4^	3.24 × 10^−7^	−533	−476
	120	2.07 × 10^5^	1.26 × 10^−7^	−593	−599
316L	0.25	3.68 × 10^4^	7.08 × 10^−7^	−326	−240
	24	3.01 × 10^4^	8.68 × 10^−7^	−325	−244
	48	2.89 × 10^4^	9.02 × 10^−7^	−319	−241
	120	3.24 × 10^4^	8.05 × 10^−7^	−327	−215

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
