# Peer review of "Production, Mechanical Properties and Biomedical Characterization of ZrTi-Based Bulk Metallic Glasses in Comparison with 316L Stainless Steel and Ti6Al4V Alloy"

_materials, 2021, doi:10.3390/ma15010252_

Round 1

Reviewer 1 Report

In this paper, the authors give us the production, mechanical properties and biomedical characterization of ZrTi-based Bulk Metallic Glasses, which is rapidly cooled, in comparison with 316L Stainless Steel and Ti6Al4V Alloy by using SEM, EDX methods, Nanoindentation test, Scratch tester, Electrochemical investigations and some biocompatibility testing. The manuscript content is new and attractive to readers. The whole manuscript is well structured and scientifically presented. Some aspects should be taken into account, as following:

  • The materials and methods section is quite detail but some chemical compound is not in the right typing. Page 3 line 142-143.
  • The sample preparation for scratch test should be provided in the section 2.3. 
  • The authors give the reader about production of three metallic glasses (Zr40Ti15Cu10Ni10Be25, Zr50Ti5Cu10Ni10Be25 and Zr40Ti15Cu10Ni5Si5Be25 (at.%) alloys), but there are no confirmation of amorphous state by XRD or TEM. Evidences for amorphous state and crystalline phases should to be added.
  • Fig 2a gives us the hardness for phases A and B and matrix, where are the phases in Fig 1. What is the state of matrix, phase A and phase B? Also error bar should be added for Fig 2a.
  • The corrosion part includes a bunch of information so it should have explanation in each time mode of exposure to SBF instead of putting these explanation in the end of this part. It will be easier to follow. The remaining parts in this section is adequate and have sufficient comment for each result.

Some latest references about metallic glasses and amorphous biometals should be cited:

  • https://doi.org/10.1002/14356007.a16_335.pub2 
  • https://doi.org/10.3390/met10020203 

Author Response

Dear Reviewer,

thank you very much for all your comments and suggestions, which significantly improve the quality of our article. Below you can find our responses to your comments. I hope that our explanations are satisfactory.

Sincerely yours,

Mariusz Hasiak

----------------------------------------------------------------

  • The materials and methods section is quite detail but some chemical compound is not in the right typing. Page 3 line 142-143.

The chemical compounds typing was corrected.

  • The sample preparation for scratch test should be provided in the section 2.3. 

The following sentence about samples’ preparation was added to the manuscript (section 2.3): “All samples were grounded with 400 - 2000 grit sandpaper and then polished with 6 - 0.25 μm diamond suspensions, followed by the final polishing step with 0.05 μm colloidal silica.”

  • The authors give the reader about production of three metallic glasses (Zr40Ti15Cu10Ni10Be25, Zr50Ti5Cu10Ni10Be25 and Zr40Ti15Cu10Ni5Si5Be25 (at.%) alloys), but there are no confirmation of amorphous state by XRD or TEM. Evidences for amorphous state and crystalline phases should to be added.

The XRD investigations were performed for all investigated materials. These studies clearly confirmed the glassy nature of the produced materials. Similar compositions were also investigated by other authors, which also confirmed our results (see e.g. refs 18, 22, 23). Therefore, on the basis of these examinations as well as SEM observations we used the term “amorphous matrix”. Moreover, detailed results on microstructure examinations, including XRD data, are currently under review in another paper.

  • Fig 2a gives us the hardness for phases A and B and matrix, where are the phases in Fig 1. What is the state of matrix, phase A and phase B? Also error bar should be added for Fig 2a.

In the investigated Zr-based bulk metallic glasses phase A corresponds to the amorphous matrix (gray matrix in Fig. 1), which dominates in whole samples’ volume. Phases B and C correspond to crystalline precipitations (dark and bright spots in Fig. 1). Due to microscale precipitations, distinguish of hardness between these phases is unequivocal. Statistical analysis of the obtained results shows that measurements errors are insignificant.

  • The corrosion part includes a bunch of information so it should have explanation in each time mode of exposure to SBF instead of putting these explanation in the end of this part. It will be easier to follow. The remaining parts in this section is adequate and have sufficient comment for each result.

From the corrosion investigation point of view, the structure of the presented results used in our paper (3.4 Corrosion section) is the standard one for this type of studies. This part was consulted with experts in corrosion science.

  • Some latest references about metallic glasses and amorphous biometals should be cited:

https://doi.org/10.1002/14356007.a16_335.pub2 

Reference 30 (published in 2017) in our manuscript written by the same authors (Challapalli Suryanarayana, Akihisa Inoue) covers similar information as it is presented in older book https://doi.org/10.1002/14356007.a16_335.pub2 (Suryanarayana, C. and Inoue, A. (2012). Metallic Glasses. In Ullmann's Encyclopedia of Industrial Chemistry, (Ed.))

https://doi.org/10.3390/met10020203 

The mentioned above reference was added to the manuscript.

Reviewer 2 Report

minor

Author Response

Dear Reviewer,

thank you very much for all your comments and suggestions, which significantly improve the quality of our article. Below you can find our responses to your comments. I hope that our explanations are satisfactory.

Sincerely yours,

Mariusz Hasiak

--------------------------------------------------------

  • The title of the manuscript is too long and I would like to recommend shortening it a little. 2-4 words reducing will be enough.

The title of the manuscript was changed according to the reviewer’s suggestion.

  • I would definitely recommend adding XRD analysis to the manuscript to show metallic glasses instead of the amorphous phase.

The XRD investigations were performed for all investigated materials. These studies clearly confirmed the glassy nature of the produced materials. Similar compositions were also investigated by other authors, which also confirmed our results (see e.g. refs 18, 22, 23). Therefore, on the basis of these examinations as well as SEM observations we used the term “amorphous matrix”. Moreover, detailed results on microstructure examinations, including XRD data, are currently under review in another paper.

  • The composition of the studied alloys is pretty close to high-entropy alloys, thus I would like to see several sentences in the manuscript regarding differences between these materials.

The chemical composition of investigated materials presented in our paper is similar to the composition of high-entropy alloys but the internal structure and physicochemical properties are significantly different. The comparison between these groups of materials is not valid due to the subject of the manuscript.

  • Such a big difference in COF values depending on the scratch length looks very confusing. Could you explain it?

The increase in COF  observed in the recorded curves is related to material build-up during scratching. The material build-up on the sides of the indenter increased the contact surface without bearing the load.

  • I would like to propose adding elemental mapping in elemental contrast after corrosion tests.

The changes on the samples’ surface are insignificant and elemental mapping will not give new information about the materials. The corrosion section was consulted with an expert in corrosion science.

  • The review of recently published articles should be expanded by the work [1], devoted to similar studies.

In our opinion, the results presented in work [1] (K. V. Smyrnova, Alexander D. Pogrebnjak and L. G. Kassenova, Structural Features and Properties of Biocompatible Ti-Based Alloys with β-Stabilizing Elements, https://doi.org/10.1007/978-981-13-6133-3_31) are not similar to those presented in our manuscript. The paper entitled  “Production, Mechanical Properties and Biomedical Characterization of ZrTi-based Bulk Metallic Glasses in Comparison with 316L Stainless Steel and Ti6Al4V Alloy” considers investigations of Zr-based bulk metallic glasses whereas the paper mentioned by the reviewer devotes to the crystalline Ti-based alloys.

Reviewer 3 Report

It is an excellent contribution, well organized and the conclusions addressed are fully supported.

Comments:

  1. Line 200: Be element is missing in the sentence.
  2. Add uncertainties in Table 1.
  3. Error bars in Figs. 5 and 8 are not fully visible.

Author Response

Dear Reviewer,

thank you very much for all your comments and suggestions, which significantly improve the quality of our article. Below you can find our responses to your comments. I hope that our explanations are satisfactory.

Sincerely yours,

Mariusz Hasiak

----------------------------------------------------------------

  • Line 200: Be element is missing in the sentence.

The correction was done according to the reviewer’s suggestion.

  • Add uncertainties in Table 1.

The errors for the potentiodynamic investigations are very small and can be neglected.

  • Error bars in Figs. 5 and 8 are not fully visible.

The correction was done according to the reviewer’s suggestion.

Round 2

Reviewer 1 Report

The revised manuscript meet all the need. I can recommend this manuscript to publication.

This manuscript is a resubmission of an earlier submission. The following is a list of the peer review reports and author responses from that submission.